# Effects of risperidone on amino acid metabolism, glucose, and kidney function in healthy adults: A pilot randomized controlled trial

Megumi Oshima[1], Tadashi Toyama[1,2], Yusuke Nakade[3], Sakae Miyagi[4], Hisayuki Ogura[1], Shiori Nakagawa[1], Takahiro Yuasa[1], Akihiko Koshino[1], Keisuke Horikoshi[1], Taichiro Minami[1], Keisuke Sako[1], Shunsuke Tsuge[1], Akira Tamai[1], Ryo Nishioka[1], Taro Miyagawa[1], Kiyoaki Ito[1], Shinji Kitajima[1], Ichiro Mizushima[1], Akinori Hara[1], Norihiko Sakai[1], Miho Shimizu[1], Toshiaki Tokumaru[5], Makoto Tsubomoto[6], Mitsuru Kikuchi[6,7], Masashi Kinoshita[8], Mitsutoshi Nakada[8], Masashi Mita[9], Yasunori Iwata[1]*, Takashi Wada[1]

**1** Department of Nephrology and Rheumatology, Kanazawa University, Kanazawa, Japan, **2** Department of Nephrology, Faculty of Medical Sciences, University of Fukui, Fukui, Japan, **3** Department of Clinical Laboratory, Kanazawa University, Kanazawa, Japan, **4** Innovative Clinical Research Center, Kanazawa University, Kanazawa, Japan, **5** Department of Nutrition, Kanazawa University Hospital, Kanazawa, Japan, **6** Department of Psychiatry and Behavioral Science, Kanazawa University Graduate School of Medical Sciences, Kanazawa, Japan, **7** Research Center for Child Mental Development, Kanazawa University, Kanazawa, Japan, **8** Department of Neurosurgery, Kanazawa University, Kanazawa, Japan, **9** KAGAMI Inc., Ibaraki, Osaka, Japan

* iwatay@staff.kanazawa-u.ac.jp

## Abstract

d-serine administration prevents kidney damage in murine models of acute kidney injury, and risperidone inhibits the activity of d-amino acid oxidase, which regulate plasma d-amino acid levels. This pilot randomized controlled trial investigated the effects of risperidone on glucose, amino acid metabolism, and kidney function in healthy adults. Healthy adults with a homeostasis model assessment of insulin resistance (HOMA-IR) of ≥ 1.6 and estimated glomerular filtration rate (eGFR) of ≥ 60 mL/min/1.73m² were randomly assigned to the risperidone and control groups. The risperidone group received 0.5 mg/day risperidone for 4 days. The primary outcome was mean change in HOMA-IR on day 5, and the secondary outcomes were changes in d-amino acid levels, eGFR, and urinary albumin. Seven participants were randomized to the risperidone and control groups. The changes in HOMA-IR, eGFR, and urinary albumin on day 5 were not significantly different between the two groups (all p > 0.05). Mean changes in plasma d-serine level and urinary d-serine/creatinine ratio were significantly higher in the risperidone group than in the control group (0.2 vs. −0.3 nmol/mL, p = 0.03 and 38.2 vs. −25.8 nmol/mL, p = 0.01, respectively). Short-term risperidone affects d-serine metabolism without instigating acute adverse effects on kidney or glucose homeostasis in healthy individuals.

**Data availability statement:** The dataset of this study is available from the Figshare repository at the following DOI: https://doi.org/10.6084/m9.figshare.30000049.v1.

**Funding:** This study was supported by KAGAMI INC. MM is the founder and CEO of KAGAMI INC., a startup company working on chiral amino acid analysis and research for medical applications. The funder provided blinded amino acid level measurements, but had no role in study design, data collection and analysis. MO is supported by a grant from the 2021 public offering research on clinical research by Kanazawa University Hospital and Initiative for Realizing Diversity in the Research Environment, MEXT. Other authors did not receive any specific grant from funding agencies in the public, commercial, or not-for-profit sectors.

**Competing interests:** MM is the founder and CEO of KAGAMI INC. This does not alter our adherence to PLOS ONE policies on sharing data and materials.

**Abbreviations:** CKD, chronic kidney disease; eGFR, estimated glomerular filtration rate; HOMA-IR, homeostasis model assessment of insulin resistance; UACR, urine albumin to creatinine ratio; rRNA, ribosomal ribonucleic acid.

**Clinical Trial Registry number:** This study was registered with the Japan Registry for Clinical Trials (jRCTs041210165).

## Introduction

Chronic kidney disease (CKD), which impacts more than 10% of the global population, is associated with severe outcomes, including end-stage kidney disease and cardiovascular complications [1]. Despite the demonstrated efficacy of several therapeutic agents, such as renin-angiotensin system inhibitors and sodium-glucose cotransporter-2 inhibitors, in slowing CKD progression, the residual risk for disease progression remains significant [2,3]. Thus, the development of novel therapeutic approaches are urgently needed to mitigate the risk and progression of CKD in the general population.

The renoprotective roles of d-amino acids, such as d-serine and d-alanine, have been demonstrated in several recent in vitro and in vivo studies [4–6]. For example, the oral administration of d-serine and d-alanine alleviated kidney damage in a murine model of acute kidney injury induced by ischemia/reperfusion [4,5], whereas elevated d-serine levels were associated with tissue remodeling in a nephrectomy model [6]. Clinical studies have also found that the plasma d-serine and d-alanine concentrations are higher in patients with CKD than in healthy individuals [7–9], suggesting their potential as biomarkers and therapeutic targets in kidney diseases.

Risperidone, an atypical antipsychotic agent widely used for the treatment of schizophrenia, exerts its primary effects through the antagonism of D2 dopamine and 5-HT2A serotonin receptors [10]. Additionally, risperidone also inhibits d-amino acid oxidase, which degrades d-serine and d-alanine [11–13]. In a previous observational study of patients in routine clinical care, we reported that risperidone use was associated with a reduced risk of decline in kidney function in patients with schizophrenia [14], raising the possibility that the effects of risperidone on d-amino acid metabolism might be a contributor to its protective effect against kidney injury.

Several studies have investigated the effects of antipsychotic medications on kidney function with mixed results. While some retrospective studies suggest that atypical antipsychotics, including risperidone, may not increase the risk of acute kidney injury compared to typical antipsychotics [15,16], other population-based studies have reported increased risks in elderly patients [17]. The mechanisms underlying these effects remain unclear, but may involve direct nephrotoxic effects or indirect effects through metabolic changes.

Despite the evidence supporting the renoprotective effect of risperidone, its impact on the amino acid profile of humans remains unclear. Additionally, the interplay of risperidone with kidney function and glucose metabolism has not been well-studied in individuals without psychiatric conditions. Therefore, we conducted a pilot randomized controlled trial to investigate the impact of short-term risperidone administration on the amino acid profile, kidney function, and glucose metabolism in healthy adults. Microbiome analysis was included because gut microbiome can metabolize D-amino acids, and we aimed to control for potential confounding effects of microbial D-amino acid

metabolism on our study endpoints. Our findings offer insights into the pharmacological effects of risperidone on d-serine and d-alanine levels, laying the foundation for novel therapeutic strategies targeting d-amino acid in patients with CKD.

## Materials and methods

### Study design and participants

This pilot, open-label, randomized controlled trial evaluated the efficacy of administration of risperidone for 4 days in healthy volunteers. The study protocol (S1 Protocol and S1 Checklist) was approved by the Medical Ethics Committee of Kanazawa University (approval no: 2021-004) and was registered at jRCT (jRCTs041210165). The study was conducted in accordance with the Declaration of Helsinki. The participants were recruited between March 22, 2022 and July 20, 2023. Written informed consent was obtained from all participants prior to enrollment.

The study enrolled healthy adults aged 20–65 years who met the following criteria: 1) no history of psychiatric or neurologic disorders, 2) fasting blood glucose levels < 140 mg/dL, 3) homeostasis model assessment of insulin resistance (HOMA-IR) ≥ 1.6, 4) estimated glomerular filtration rate (eGFR) ≥ 60 mL/min/1.73 m$^2$, 5) both aspartate and alanine transaminase levels < 30 IU/L, 6) no history of arrhythmias or congenital long QT syndrome, 7) no history of cancer within the past five years, and 8) presence of bowel movements at least once every four days, regardless of laxative use. The glucose cutoff was chosen to exclude participants with overt diabetes while including those with potential insulin resistance, as measured by HOMA-IR.

Participants fulfilling the following criteria were excluded from the study: 1) any medically diagnosed condition currently under treatment, 2) history of hypersensitivity to risperidone, paliperidone, or their components, 3) pregnancy or breastfeeding, 4) participation in another clinical study within the last 3 months prior to the initiation of risperidone administration, 5) deemed unsuitable for study participation for any reason by the principal or sub-investigator, and 6) currently holding a faculty or staff position of the department in which the principal investigator is affiliated.

Eligible participants were randomized at a 1:1 ratio to the risperidone and control groups within 30 days of the study enrollment (S1 Fig). Randomization was centrally performed in the registration center using minimization, with age and sex as allocation adjustment factors. A sample size estimation was not performed due to the absence of prior data on effect size. This pilot study was designed to generate preliminary data for future power calculations.

### Interventions

Participants in the risperidone group were administered 0.5 mg risperidone (Risperdal® OD, 0.5 mg) once daily at bedtime for 4 days. The lowest available dosage of 0.5 mg/day was chosen to prioritize participant safety. The 4-day treatment duration was chosen based on pharmacokinetic studies showing that steady-state levels are achieved within 3–4 days, allowing us to observe steady-state effects while minimizing exposure in healthy volunteers. To ensure compliance, participants in the risperidone group were instructed to return any remaining tablets at the end of the study for collection.

The study was conducted in an outpatient setting, and no participants was hospitalized during the intervention. Regardless of the group assignment, all participants visited the study site on days 1, 5, and 9 after study initiation. To minimize interference with the gut microbiota, participants were asked to restrict the consumption of foods containing probiotics, such as yogurt, fermented beverages, natto, miso, pickles, and kimchi, starting one week before the study initiation and continuing throughout the follow-up period. Participants were also restricted from using antimicrobials and *Lactobacillus* preparations. Furthermore, participants were advised to avoid activities requiring alertness, such as driving or operating hazardous machinery, and to abstain from alcohol to reduce the risk of adverse events during the risperidone administration period.

### Laboratory measurements

Participants underwent blood tests after fasting or at least 6 hours after meals and early morning urine tests at baseline and on days 5 and 9 after study initiation. Blood parameters that were assessed included leukocytes, erythrocytes, and

platelet counts; hemoglobin; hematocrit; total protein; serum albumin; aspartate and alanine aminotransferases; blood urea nitrogen; creatinine; uric acid; sodium; potassium; calcium; inorganic phosphorus; fasting blood glucose; hemoglobin A1c; insulin; C-peptide; total cholesterol; high-density lipoprotein cholesterol; and triglycerides.

The following equation proposed by the Japanese Society of Nephrology was used to calculate eGFR: eGFR (mL/min/1.73 m$^2$) = 194 × serum creatinine (mg/dL) $^{-1.094}$ × age (years)$^{-0.287}$ × 0.739 (if female) [18]. The urine albumin to creatinine ratio (UACR) was also assessed. All blood and urinary parameters were measured by an independent laboratory at BML Inc. (Tokyo, Japan) using automated clinical testing technology. The results of all urinary parameters were normalized to urine creatinine to correct for differences in concentrations uniquely related to the hydration status or the urine volume of the participant [19].

Plasma and urinary levels of dl-serine and alanine at baseline and on day 5 were determined using a two-dimensional high-performance liquid chromatography system (DASH 27B3X00322000001) by KAGAMI INC. (Osaka, Japan), according to a previously described protocol with modifications [4,5,8].

Intestinal bacterial flora analysis was conducted based on 16S ribosomal ribonucleic acid (rRNA) sequencing of fecal samples using QIIME2 (version 2021.2) by Takara Bio's Biomedical Center (Shiga, Japan). Fecal samples were collected by participants at home using a collection kit (FS-0017, TechnoSuruga Lab. Co., Ltd., Shizuoka, Japan) at baseline and on day 5. Microbiome analysis included alpha diversity, such as number of operational taxonomic units, Chao-1 index, and Shannon index, and beta diversity, such as weighted Unifrac and Bray-Curtis distances, and relative abundance, with a sampling depth of 10,000 sequences.

### Study endpoints

The primary study endpoint was the mean change in HOMA-IR from baseline to 5 days between groups. HOMA-IR was selected as the primary endpoint because risperidone has the potential to increase glucose levels and insulin resistance [20,21], and we first aimed to assess the metabolic safety profile of risperidone in healthy individuals. Intention-to-treat analyses were performed according to the randomly assigned groups. The secondary endpoints included the mean changes in the following parameters from baseline to 5 days between groups: plasma and urine dl-serine and dl-alanine levels, eGFR, blood pressure, body mass index, blood glucose, hemoglobin A1c, insulin, C-peptide, and UACR. Safety was monitored by assessing adverse events, medical interviews, and the review of laboratory values during the study period.

### Statistical analysis

For baseline participant characteristics, continuous variables were presented as means with standard deviations and categorical variables were presented as numbers with percentages. For the primary and secondary endpoints, Student's *t*-test was used to compare mean changes in clinical and laboratory parameters from baseline to days 5 and 9 between the risperidone and control groups. Given the small sample size, Mann-Whitney U test was also performed as a non-parametric analysis to complement the parametric tests. To address within-subject correlations, we performed additional paired t-tests comparing each follow-up time point (days 5 and 9) to baseline within each treatment group. All analyses were performed using Stata version 18 (Stata Corp, College Station, TX). A two-sided p value of <0.05 was considered statistically significant.

## Results

### Baseline characteristics of the participants

Between March and November 2022, a total of 8 participants were randomized to the risperidone and control groups (n = 4 participants per group), 1 participant declined to complete the study, and 7 completed the protocol treatment (Fig 1). In the

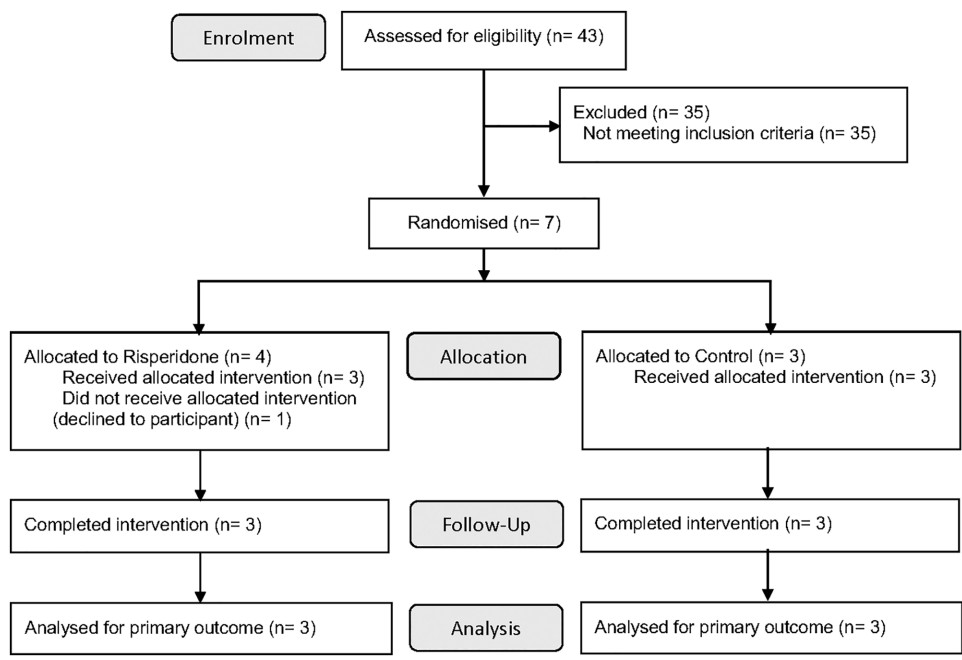

**Fig 1. Flowchart of the study.**

risperidone and control groups, the mean participant age was 35 and 29 years, 1 (33%) and 2 (50%) of the participants were male, the mean HOMA-IR was 2.0 and 1.6, and the mean eGFR was 90 and 92 mL/min/1.73 m², respectively (Table 1). The baseline participant characteristics were comparable between the two groups (all p > 0.10), except for body mass index, which was likely to be higher in the risperidone group than in the control group (p = 0.03).

## Comparison of changes in dl-amino acid levels

At baseline, the mean (± SD) plasma d-serine levels were 1.8 ± 0.3 nmol/mL and 1.9 ± 0.1 nmol/mL in the risperidone and control groups, respectively. The mean plasma d-serine level increased by 0.24 ± 0.36 nmol/mL on day 5 compared to the baseline in the risperidone group and decreased by −0.32 ± 0.13 nmol/mL during the same time period in the control group (Fig 2A and S1 Table). The mean plasma d-serine level was 0.56 (95% CI 0.07–1.05) nmol/mL higher in the risperidone group than in the control group on day 5 after the study initiation (p = 0.03). Similarly, the mean change in urinary d-serine-to-creatinine ratio was significantly higher in the risperidone group than in the control group (38.2 ± 19.8 and −25.8 ± 23.4 mmol/g Cr; respectively; between-group difference: 64.0 mmol/g Cr, 95% CI 20.8–107.2; p = 0.01) compared to the control group. The significant differences in plasma d-serine levels and urinary d-serine-to-creatinine ratio between groups remained consistent when analyzed using Mann-Whitney U tests (S1 Table).

The mean changes in plasma d-alanine level and urinary d-alanine-to-creatinine ratio were numerically higher in the risperidone group than in the control group, with no statistical difference (0.54 vs. −0.31 nmol/mL, p = 0.25 and 24.98 vs. −9.42 nmol/mL, p = 0.20, respectively) (Fig 2B and S1 Table). Conversely, no significant changes were observed in plasma and urinary l-serine and l-alanine levels during the follow-up period and these parameters did not significantly differ between the two groups (Fig 2 and S1 Table). Within-group paired comparisons showed that the control group had a significant decrease in d-serine from baseline to day 5 (p = 0.02), while other amino acids showed no significant within-group changes across any group (all p > 0.07).

**Table 1. Baseline participant characteristics in the risperidone and control groups.**

| | Risperidone (n = 3) | Control (n = 4) | p |
|---|---|---|---|
| Age (years) | 35 (12) | 29 (8) | 0.49 |
| Male | 1 (33) | 2 (50) | 0.66 |
| Current smoking | 0 (0) | 0 (0) | – |
| Drinking habit | 1 (33) | 1 (25) | 0.81 |
| Body weight (kg) | 66 (9) | 61 (7) | 0.41 |
| Body mass index (kg/m$^2$) | 23.8 (1.4) | 21.2 (0.8) | 0.03 |
| Systolic blood pressure (mmHg) | 105 (6) | 113 (16) | 0.45 |
| Diastolic blood pressure (mmHg) | 71 (9) | 74 (5) | 0.69 |
| Aspartate aminotransferase (U/L) | 16 (4) | 19 (6) | 0.62 |
| Alanine aminotransferase (U/L) | 12 (6) | 16 (7) | 0.50 |
| eGFR (mL/min/1.73 m$^2$) | 90 (20) | 92 (13) | 0.89 |
| Uric acid (mg/dL) | 4.6 (1.0) | 6.3 (2.0) | 0.25 |
| Fasting blood glucose (mg/dL) | 92 (9) | 90 (4) | 0.67 |
| Hemoglobin A1c (%) | 4.9 (0.1) | 5.0 (0.2) | 0.61 |
| Insulin (μU/mL) | 8.9 (2.0) | 7.3 (2.9) | 0.44 |
| C peptide (ng/mL) | 1.9 (0.4) | 1.6 (0.5) | 0.33 |
| HOMA-IR | 2.0 (0.4) | 1.6 (0.6) | 0.36 |
| HOMA-β (%) | 119 (57) | 100 (43) | 0.64 |
| Total cholesterol (mg/dL) | 227 (35) | 183 (23) | 0.10 |
| HDL-cholesterol (mg/dL) | 66 (21) | 67 (16) | 0.97 |
| Triglycerides (mg/dL) | 82 (19) | 73 (23) | 0.60 |
| Urine albumin/Cr (mg/g Cr) | 2.9 (1.1) | 4.7 (2.9) | 0.38 |
| **Plasma amino acid** | | | |
| d-serine (nmol/mL) | 1.8 (0.3) | 1.9 (0.1) | 0.52 |
| l-serine (nmol/mL) | 120.4 (18.4) | 133.3 (17.8) | 0.39 |
| d/l-serine (%) | 1.5 (0.5) | 1.4 (0.2) | 0.77 |
| d-alanine (nmol/mL) | 1.0 (0.2) | 0.8 (0.7) | 0.75 |
| l-alanine (nmol/mL) | 354.3 (50.4) | 288.5 (59.8) | 0.19 |
| d/l-alanine (%) | 0.3 (0.1) | 0.3 (0.2) | 0.96 |
| **Urine amino acid** | | | |
| d-serine/Cr (mmol/g Cr) | 123.4 (10.0) | 136.6 (18.3) | 0.31 |
| l-serine/Cr (mmol/g Cr) | 118.9 (53.3) | 256.6 (252.1) | 0.40 |
| d-alanine/Cr (mmol/g Cr) | 27.1 (9.5) | 23.7 (19.6) | 0.80 |
| l-alanine/Cr (mmol/g Cr) | 142.3 (75.5) | 113.7 (46.3) | 0.56 |

Abbreviations; eGFR, estimated glomerular filtration rate; HOMA-IR, Homeostatic model assessment of insulin resistance; HOMA-β, Homeostatic model assessment of beta cell function; HDL, high density lipoprotein; Cr, creatinine.

Data are described as mean (standard deviation) for continuous variables or n (%) for categorical variables. Statistical comparisons were performed using Student's *t*-test for continuous variables and Fisher's exact test for categorical variables.

## Effects of risperidone on clinical and laboratory parameters

The mean change in HOMA-IR was not statistically significantly different between the risperidone and control groups during the treatment period (0.88 ± 0.58 and 0.42 ± 1.00, p = 0.51) (Fig 3 and Table 2). The change in mean blood glucose from the baseline to day 5 after study initiation was marginally higher in the risperidone group than in the control group by

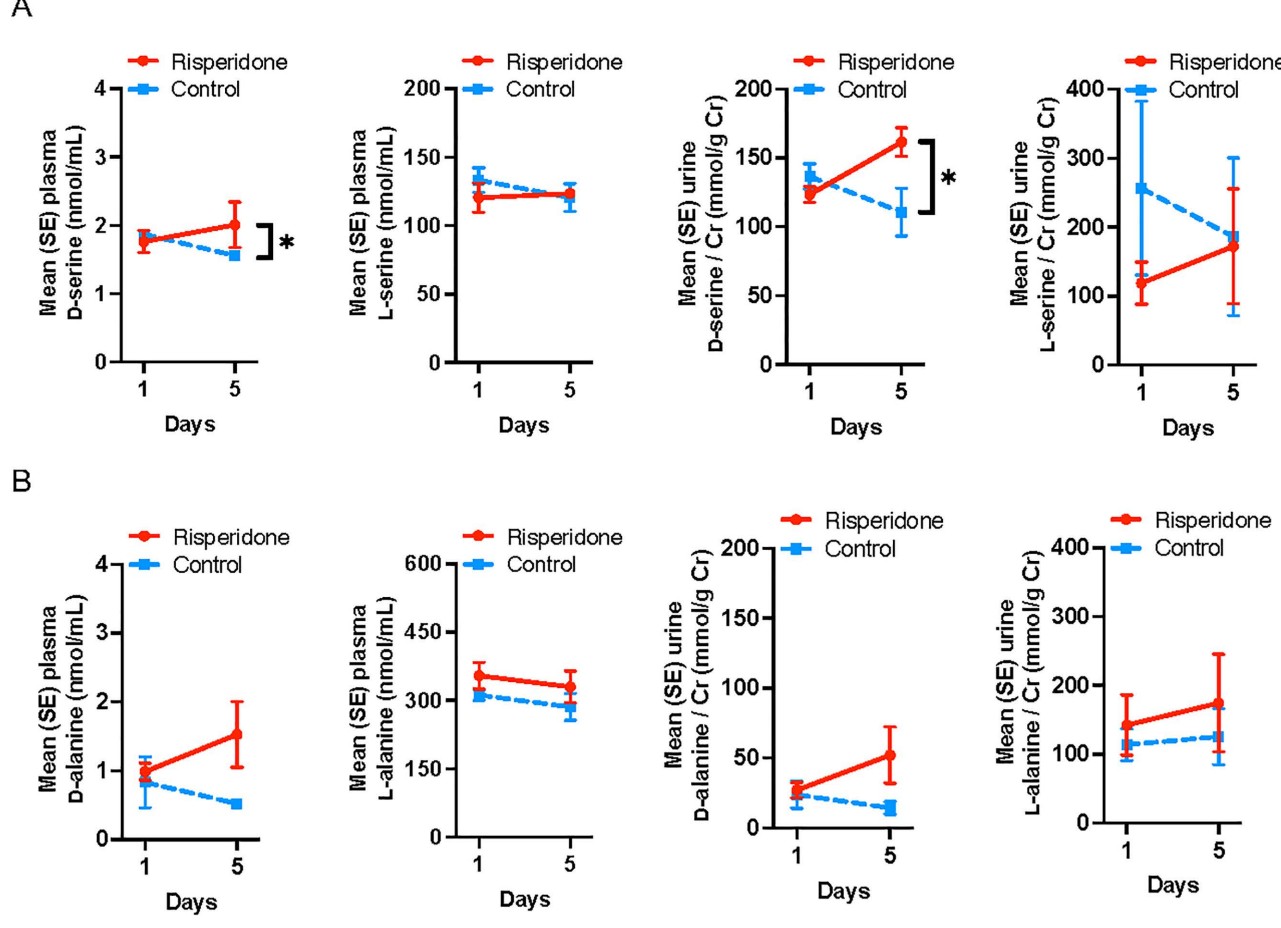

\* P < 0.05

**Fig 2. Mean plasma and urine dl-serine (A) and dl-alanine (B) levels during follow-up in the risperidone and control groups.**

Student's t-test (4.0±2.0 vs. −1.3±3.0 mg/dL, respectively; p=0.048), however this difference was not significant when analyzed using the Mann-Whitney U test (p=0.07). The mean change in eGFR did not differ between the risperidone and control groups (1.0±8.7 and −0.4±15.6 mL/min/1.73 m$^2$, respectively; p=0.90). The changes in mean blood pressure, body mass index, hemoglobin A1c, insulin, C-peptide, and UACR were not statistically significantly different between the two groups (Table 2). All other parameters except fasting blood glucose showed consistent results between parametric and non-parametric analyses. Within-group analysis revealed that each group showed no significant changes in metabolic parameters at either time point (all p>0.07) (Table 2).

Exploratory analysis to assess the effects of risperidone on additional clinical and laboratory parameters (S2 Table) revealed no significant changes in body weight, HOMA for β-cell function, aspartate and alanine aminotransferase, uric acid, total and high-density lipoprotein cholesterol, and triglycerides after 4 days of risperidone administration.

### Effects of risperidone on the fecal microbiome

In the 16S rRNA analysis, no significant differences were observed in alpha diversity (species richness) metrics of the fecal microbiome on day 5 compared to the baseline in both risperidone and control group (paired *t*-test, p>0.10) (S2 Fig).

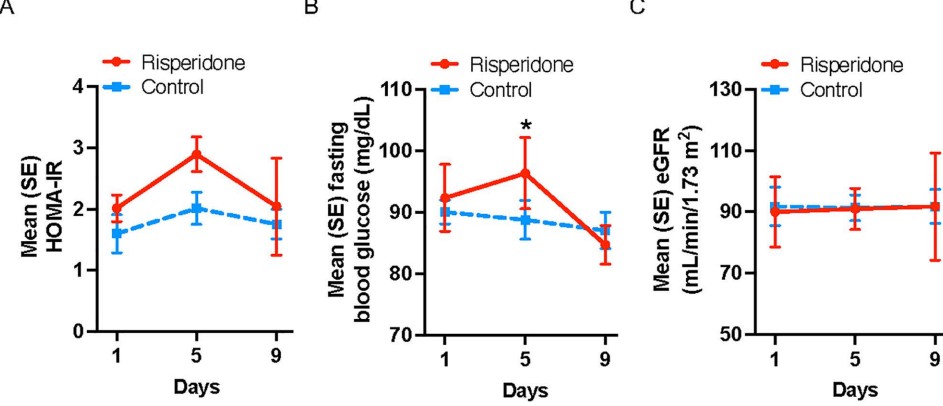

* P < 0.05

**Fig 3. Mean homeostatic model assessment of insulin resistance (HOMA-IR) (A), fasting blood glucose (B), and estimated glomerular filtration rate (eGFR) (C) levels during follow-up in the risperidone and control groups.**

**Table 2. Mean changes in clinical and laboratory parameters in the risperidone and control groups.**

| | | Mean (SD) change from baseline | | Between groups | | Within-group p | |
|---|---|---|---|---|---|---|---|
| | Day | Risperidone (n=3) | Control (n=4) | p* | p** | Risperidone | Control |
| HOMA-IR | 5 | 0.88 (0.58) | 0.42 (1.0) | 0.51 | 0.29 | 0.12 | 0.47 |
| | 9 | 0.03 (1.25) | 0.15 (0.88) | 0.88 | 1.00 | 0.97 | 0.74 |
| Systolic blood pressure (mmHg) | 5 | 7.7 (18.0) | −3.3 (7.7) | 0.32 | 0.29 | 0.54 | 0.46 |
| | 9 | 8.0 (19.5) | 1.8 (15.3) | 0.65 | 0.48 | 0.55 | 0.83 |
| Diastolic blood pressure (mmHg) | 5 | −2.3 (13.7) | −3.5 (2.5) | 0.87 | 0.48 | 0.80 | 0.07 |
| | 9 | 1.0 (8.9) | −2.3 (8.7) | 0.65 | 0.48 | 0.86 | 0.64 |
| Body mass index (kg/m²) | 5 | 0.1 (0.2) | −0.2 (0.2) | 0.10 | 0.08 | 0.33 | 0.20 |
| | 9 | −0.03 (0.2) | −0.06 (0.2) | 0.87 | 1.00 | 0.84 | 0.63 |
| eGFR (mL/min/1.73 m²) | 5 | 1.0 (8.7) | −0.4 (15.6) | 0.90 | 0.72 | 0.87 | 0.96 |
| | 9 | 1.8 (10.7) | 0.02 (2.8) | 0.76 | 0.72 | 0.80 | 0.99 |
| Fasting blood glucose (mg/dL) | 5 | 4.0 (2.0) | −1.3 (3.0) | 0.048 | 0.07 | 0.07 | 0.46 |
| | 9 | −7.7 (4.7) | −3.0 (3.8) | 0.21 | 0.21 | 0.11 | 0.22 |
| Hemoglobin A1c (%) | 5 | 0.00 (0.10) | 0.08 (0.05) | 0.24 | 0.29 | 1.00 | 0.18 |
| | 9 | 0.07 (0.06) | 0.05 (0.06) | 0.72 | 1.00 | 0.18 | 0.42 |
| Insulin (µU/mL) | 5 | 3.2 (2.1) | 1.9 (4.0) | 0.62 | 0.48 | 0.11 | 0.65 |
| | 9 | 1.2 (5.8) | 0.9 (3.8) | 0.95 | 1.00 | 0.75 | 0.67 |
| C peptide (ng/mL) | 5 | 0.90 (0.95) | 0.20 (0.59) | 0.28 | 0.16 | 0.24 | 0.94 |
| | 9 | 0.03 (0.67) | 0.03 (0.59) | 0.99 | 0.86 | 0.94 | 0.53 |
| Urine albumin/Cr (mg/g Cr) | 5 | −0.1 (1.0) | −0.3 (2.4) | 0.92 | 0.72 | 0.86 | 0.83 |
| | 9 | −0.4 (1.1) | −1.4 (2.2) | 0.50 | 0.48 | 0.63 | 0.29 |

Abbreviations; HOMA-IR, Homeostatic model assessment of insulin resistance; eGFR, estimated glomerular filtration rate; Cr, creatinine. P* values were calculated using Student's t-test and p** values were calculated using Mann-Whitney U test as non-parametric analysis between groups. Within-group p values were calculated using paired t-tests comparing each time point to baseline within each group.

For beta diversity (overall structural similarity and variation), the weighted Unifrac distances were visually unchanged, and the Gray-Curtis distances were similar (p = 0.81) between the risperidone and control groups during the treatment period (S2 Fig). Phylum-level analysis revealed that both groups were dominated by Firmicutes and Bacteroidetes, accounting for over 80% of the microbiome composition at baseline and no significant changes were observed in the composition at day 5 in either group (S2 Fig and S3 Table).

## Safety

All adverse events observed during the study are presented in Table 3. All participants reported a total of eight non-serious adverse events. In the risperidone group, two participants experienced somnolence, and one participant experienced abdominal distension. One participant in each group reported blood pressure of > 140/90 mmHg. No participant in the risperidone group reported hypotension or dizziness, and no serious adverse events were recorded.

## Discussion

In this open-label, randomized pilot clinical trial, we found that the short-term administration of risperidone led to an increase in plasma and urinary d-serine levels in healthy adults with normal kidney function. Additionally, risperidone did not adversely affect eGFR levels or insulin resistance. Overall, these results suggest that risperidone impacts d-serine metabolism without an acute adverse impact on kidney function or glucose homeostasis in healthy individuals.

The significant elevation in d-serine levels observed immediately after risperidone administration is a notable finding, which is likely attributable to risperidone-mediated inhibition of d-amino acid oxidase, preventing d-serine degradation [11,22]. The contribution of dietary and microbial factors to d-serine levels should also be considered. d-serine is present in various food sources, including fermented products rich in d-amino acids due to microbial fermentation by serine racemase [23–26]. Although fecal d-serine levels were not measured in the present study, we aimed to minimize confounding mediated dietary factors through the restriction of the intake of fermented foods, antibiotics, and lactic acid supplements during the study period. In addition, the lack of significant changes in gut microbiome diversity during the short treatment period suggests that the increases in d-serine levels were primarily due to direct enzyme inhibition rather than microbial effects. These findings ensured that the observed changes in d-serine were attributable to risperidone administration rather than the impact of dietary or gut microbiota-related factors.

Given the small sample size of this pilot study, the magnitude of observed effects may be more informative than statistical significance. The substantial increase in plasma d-serine levels (0.56 nmol/mL difference, 95% CI 0.07–1.05) and urinary d-serine-to-creatinine ratio (64.0 mmol/g Cr difference, 95% CI 20.8–107.2) represent clinically meaningful changes that warrant further investigation in larger studies. Similarly, the numerical increase observed in plasma d-alanine levels

**Table 3. Safety in the risperidone and control groups.**

|  | N (%) | |
|---|---|---|
|  | Risperidone (n = 3) | Control (n = 4) |
| Somnolence | 2 (67) | 0 (0) |
| Hypertension | 1 (33) | 1 (25) |
| Abdominal distension | 1 (33) | 0 (0) |
| Hypertriglyceridemia | 0 (0) | 1 (25) |
| Hypercholesterolemia | 0 (0) | 1 (25) |
| Liver injury | 0 (0) | 1 (25) |
| Hypotension | 0 (0) | 0 (0) |
| Dizziness | 0 (0) | 0 (0) |

and urinary d-alanine-to-creatinine ratio, while not reaching statistical significance, provide valuable preliminary data for effect size calculations in future trial designs.

Short-term risperidone administration did not affect kidney function in healthy individuals enrolled in our study. Few studies have investigated the efficacy and safety of risperidone on kidney function, with contradictory findings. Results from several retrospective cohort studies suggest that risperidone does not increase the risk of acute kidney injury compared to typical antipsychotics [15,16]. In a previous retrospective observational study, we also found that risperidone use was associated with a lower risk of decline in kidney function in patients with schizophrenia [14]. However, a population-based cohort study reported an increased risk of acute kidney injury-related hospitalizations in elderly patients treated with atypical antipsychotics, including risperidone [17]. The absence of acute adverse effects on kidney function in healthy participants in our study contrasts with some previous reports of antipsychotic-associated nephrotoxicity, which may be related to our study population (healthy individuals vs. patients with comorbidities) or the short duration of exposure. These discrepancies underscore the need for further studies to elucidate the kidney safety profile of risperidone particularly in vulnerable populations such as elderly individuals and patients with CKD.

We also found a slight increase in blood glucose levels during risperidone administration. Given the high variability of glucose levels influenced by diet and lifestyle, this change might not be directly related to risperidone administration. Additionally, no significant changes were observed in other glucose metabolism markers, including HOMA-IR and insulin, indicating that risperidone might have a limited impact on glycemic control during short-term use. These results align with previous studies reporting the relatively modest effect of risperidone on glucose metabolism compared to other atypical antipsychotics, which are known for their association with metabolic side effects [27,28].

This study has several limitations. First, the open-label design might have introduced performance bias due to the lack of blinding. Second, we focused exclusively on a healthy population in a pilot study with a small sample size and short follow-up period, which limited the robustness of the safety data. Any inferences should be interpreted with caution. The small sample size limits our findings and any inferences should be interpreted with caution. Third, the assumption of normality for Student's *t*-test may not be reliably verified with small sample sizes, which represents a limitation of our statistical approach. Longer-term clinical trials with larger sample sizes in patients with CKD are warranted to assess the sustained effects and safety of risperidone in this patient population. Fourth, we did not specifically control for protein intake during the study period, which may have influenced kidney parameters. Finally, the study was conducted in a Japanese population and our findings might not be generalizable to other ethnic groups with different genetic or dietary backgrounds.

## Conclusion

In conclusion, short-term administration of risperidone at 0.5 mg over 4 days effectively increased plasma d-serine concentrations in healthy participants. Although a slight increase in blood glucose levels was observed, other markers of glucose metabolism and kidney function remained stable during the study period. These findings suggest that risperidone modulates d-serine metabolism without acute adverse effects on kidney function or glucose homeostasis in healthy individuals. These findings provide a foundation for future studies exploring the therapeutic applications of risperidone in kidney disease, and the potential safety and efficacy of risperidone in modulating d-serine metabolism warrant further investigation in clinical trials including patients with CKD.

## Supporting information

**S1 Protocol. Study protocol in Japanese and Japanese to English translation.**
(PDF)

**S1 Checklist. CONSORT checklist.**
(PDF)

**S1 Fig. Study design.**
(TIFF)

**S2 Fig. Alpha (A) and beta (B) diversities of the fecal microbiome in the risperidone and control groups.**
(TIFF)

**S1 Table. Mean changes in plasma and urine dl-serine and alanine levels in the risperidone and control groups.**
Abbreviations; Cr, creatinine.
(PDF)

**S2 Table. Mean changes in clinical and laboratory parameters in the risperidone and control groups.** Abbreviations; HOMA-β, Homeostatic model assessment of beta cell function; HDL, high density lipoprotein.
(TIFF)

**S3 Table. Relative abundance of the fecal microbiome during follow-up in the risperidone and control groups.**
(TIFF)

## Acknowledgments

The authors thank the participants who took part in the study. We would like to acknowledge Yuko Oyama, Mari Shimizu, and Eri Umeda at Kanazawa University, who have assisted in sample collection and processing. We also thank Hiroshi Imoto, Eiichi Negishi, Maiko Nakane, and Shoto Ishigo in KAGAMI INC. who have contributed to the development of alanine powder formulation and chiral amino acid analysis.

## Author contributions

**Conceptualization:** Megumi Oshima, Tadashi Toyama, Yusuke Nakade, Hisayuki Ogura, Norihiko Sakai, Miho Shimizu, Makoto Tsubomoto, Mitsuru Kikuchi, Masashi Kinoshita, Mitsutoshi Nakada, Yasunori Iwata, Takashi Wada.

**Data curation:** Megumi Oshima, Masashi Mita.

**Formal analysis:** Megumi Oshima, Tadashi Toyama, Sakae Miyagi.

**Funding acquisition:** Megumi Oshima, Yasunori Iwata.

**Investigation:** Megumi Oshima, Hisayuki Ogura, Shiori Nakagawa, Akihiko Koshino, Keisuke Horikoshi, Taichiro Minami, Keisuke Sako, Shunsuke Tsuge, Akira Tamai, Ryo Nishioka, Taro Miyagawa, Kiyoaki Ito, Shinji Kitajima, Ichiro Mizushima, Akinori Hara, Norihiko Sakai, Miho Shimizu, Toshiaki Tokumaru.

**Methodology:** Megumi Oshima, Tadashi Toyama, Sakae Miyagi, Hisayuki Ogura, Makoto Tsubomoto, Yasunori Iwata, Takashi Wada.

**Project administration:** Megumi Oshima, Hisayuki Ogura, Yasunori Iwata, Takashi Wada.

**Resources:** Megumi Oshima.

**Software:** Megumi Oshima.

**Supervision:** Megumi Oshima, Tadashi Toyama, Mitsuru Kikuchi, Masashi Kinoshita, Mitsutoshi Nakada, Yasunori Iwata, Takashi Wada.

**Validation:** Megumi Oshima.

**Visualization:** Megumi Oshima.

**Writing – original draft:** Megumi Oshima.

**Writing – review & editing:** Megumi Oshima, Tadashi Toyama, Yusuke Nakade, Sakae Miyagi, Hisayuki Ogura, Shiori Nakagawa, Takahiro Yuasa, Akihiko Koshino, Keisuke Horikoshi, Taichiro Minami, Keisuke Sako, Shunsuke Tsuge, Akira Tamai, Ryo Nishioka, Taro Miyagawa, Kiyoaki Ito, Shinji Kitajima, Ichiro Mizushima, Akinori Hara, Norihiko Sakai, Miho Shimizu, Toshiaki Tokumaru, Makoto Tsubomoto, Mitsuru Kikuchi, Masashi Kinoshita, Mitsutoshi Nakada, Masashi Mita, Yasunori Iwata, Takashi Wada.

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
