## [Decision Letter · Decision Letter 0]

10 Jun 2025

Dear Dr. Iwata,

Thank you for submitting your manuscript to PLOS ONE. After careful consideration, we feel that it has merit but does not fully meet PLOS ONE’s publication criteria as it currently stands. Therefore, we invite you to submit a revised version of the manuscript that addresses the points raised during the review process.

**ACADEMIC EDITOR: **

Please address all reviewer comments point-by-point and make revisions according to the instructions below.

We look forward to receiving your revised manuscript.

Kind regards,

Kyle J Burghardt

Academic Editor

PLOS ONE

 [This study was supported by KAGAMI INC., a startup company working on chiral amino acid analysis and research for medical applications. MO is supported by a grant from the 2021 public offering research on clinical research by Kanazawa University Hospital and Initiative for Realizing Diversity in the Research Environment, MEXT. Other authors did not receive any specific grant from funding agencies in the public, commercial, or not-for-profit sectors.]. 

[MM is the founder and CEO of KAGAMI INC.].

5. In the online submission form, you indicated that [Data described in the manuscript will be considered for availability upon request to the corresponding author].

Reviewers' comments:

Reviewer's Responses to Questions

**Comments to the Author**

1. Is the manuscript technically sound, and do the data support the conclusions?

Reviewer #1: Yes

Reviewer #2: Yes

2. Has the statistical analysis been performed appropriately and rigorously?

Reviewer #1: Yes

Reviewer #2: No

3. Have the authors made all data underlying the findings in their manuscript fully available?

Reviewer #1: No

Reviewer #2: No

4. Is the manuscript presented in an intelligible fashion and written in standard English?

Reviewer #1: Yes

Reviewer #2: Yes

Reviewer #1: The authors recruited seven healthy participants for a pilot randomized controlled trial to investigate the effects of risperidone on glucose, amino acid metabolism, and kidney function. The study reported significant results for the changes in plasma d-serine levels and urinary d-serine/creatinine ratios between the risperidone and control groups.

1. It would be informative to know whether there is any difference in the baseline characters between the two groups.

2. The sample size is extremely small. The inference based on this pilot study is questionable to some extent.

3. Student’s t-test assumes normality and equal variances. How to check the normality with such a small sample?

4. With such a small sample size, the focus still seems to be on the inference or significance for the current version. However, its effect sizes might be more valuable than determining the significance.

Reviewer #2: The authors present an analysis of glucose, renal and microbiome variables in a healthy population treated with short-term risperidone therapy. The manuscript has much potential but suffers from a lack of focus, or at least rationale, for the analysis of renal, glucose and microbiome variables at the same time. If this is improved, along with a more sufficient introduction/discussion, the manuscript is likely a candidate for publication.

-I think the background could benefit from some general information and references regarding the effects of antipsychotics on kidney function to help link your two topics here. A paragraph would be plenty to highlight current evidence regarding various antipsychotics and their influence on eGFR, CrCl, etc

-there needs to be several rationale given for methodological choices: why that duration for risp treatment? why sample size even if you did not perform a power calculation? why 140 as the cutoff for glucose which doesn't appear to rule out pre-diabetes? Why the significant instruction and concern for the effect of the microbiome which isn't mentioned in the background? This becomes even more important since there is a sudden introduction of microbiome analysis in the methods. This should be in the background with rationale for its study...

-You have some instruction about dietary restrictions in this study, was protein intake a concern given its potential effects on renal labs?

-the focus of the background is on renal function but the primary endpoint was HOMA-IR? Is it possible to explain why this is the primary endpoint?

-how and when were fecal samples collected for microbiome analysis

-did between group comparisons of mean changes correct for baseline values?

-for the demographic/baseline comparisons between the groups, can you put statistical analyses to these statements of being comparable and the BMI "likely" being different? Were any identified differences controlled for in a sensitivity analysis to assess for potential effects on primary and secondary outcomes. Perhaps this should be included in the statistics of table 1?

-table 1 footnote should explain what data is (%, means, s.d., etc)

-Is it possible to utilize the supplementary material to provide in-depth data of your microbiome analysis? The presentation of microbiome results is underwhelming given the breadth of data generally obtained from microbiome analysis.

-The discussion is lacking in depth and detail, especially as it regards the microbiome analysis, why your renal findings might contradict that of other studies.

**Do you want your identity to be public for this peer review?** For information about this choice, including consent withdrawal, please see our Privacy Policy

Reviewer #1: No

Reviewer #2: No

---

## [Author Response · Author response to Decision Letter 1]

28 Aug 2025

COMMENTS TO THE AUTHOR:

Reviewer 1:

1. It would be informative to know whether there is any difference in the baseline characters between the two groups.

Response: Thank you for this important comment. We have added statistical comparisons of baseline characteristics between the two groups to the results section and Table 1 in the revised manuscript.

Results (page 10):

The baseline participant characteristics were comparable between the two groups (all p > 0.10), except for body mass index, which was likely to be higher in the risperidone group than in the control group (p = 0.03).

2. The sample size is extremely small. The inference based on this pilot study is questionable to some extent.

Response: We acknowledge that the small sample size is a significant limitation of this pilot study and have emphasized the limitation in the discussion section in the revised manuscript.

Discussion (page 15):

This study has several important limitations…. Second, we focused exclusively on a healthy population in a pilot study with a small sample size and short follow-up period, which limited the robustness of the safety data. The small sample size limits our findings and any inferences should be interpreted with caution…. Longer-term clinical trials with larger sample sizes in patients with CKD are warranted to assess the sustained effects and safety of risperidone in this patient population.

3. Student's t-test assumes normality and equal variances. How to check the normality with such a small sample?

Response: We agree with the reviewer’s comment and have added Mann-Whitney U tests as non-parametric analyses to complement Student's t-tests. The results remained consistent between parametric and non-parametric approaches, except for fasting blood glucose where the Mann-Whitney U test did not show a significant difference. We have added these results and limitation to the revised manuscript.

Methods, Statistical analysis (page 8):

….For the primary and secondary endpoints, Student’s t-test was used to compare mean changes in clinical and laboratory parameters from baseline to days 5 and 9 between the risperidone and control groups. Given the small sample size, Mann-Whitney U test was also performed as a non-parametric analysis to complement the parametric tests.

Results (page 10):

…. The significant differences in plasma d-serine levels and urinary D-serine-to-creatinine ratio between groups remained consistent when analyzed using Mann-Whitney U tests (S1 Table).

Results (page 11):

…. The mean change in fasting blood glucose was marginally higher in the risperidone group than in the control group by Student's t-test (4.0 ± 2.0 vs. −1.3 ± 3.0 mg/dL, p = 0.048), however this difference was not significant when analyzed using the Mann-Whitney U test (p = 0.07). …. All other parameters except fasting blood glucose showed consistent results between parametric and non-parametric analyses.

Discussion (page 15):

…. Third, the assumption of normality for Student's t-test may not be reliably verified with small sample sizes, which represents a limitation of our statistical approach. ….

4. With such a small sample size, the focus still seems to be on the inference or significance for the current version. However, its effect sizes might be more valuable than determining the significance.

Response: Thank you for this important suggestion. We have added discussion emphasizing the magnitude of effects rather than statistical significance in the discussion section.

Discussion (page 14):

Given the small sample size of this pilot study, the magnitude of observed effects may be more informative than statistical significance. The substantial increase in plasma D-serine levels (0.56 nmol/mL difference, 95% CI 0.07–1.05) and urinary D-serine-to-creatinine ratio (64.0 mmol/g Cr difference, 95% CI 20.8–107.2) represent clinically meaningful changes that warrant further investigation in larger studies. Similarly, the numerical increase observed in plasma D-alanine levels and urinary D-alanine-to-creatinine ratio, while not reaching statistical significance, provide valuable preliminary data for effect size calculations in future trial designs.

Reviewer 2:

1. I think the background could benefit from some general information and references regarding the effects of antipsychotics on kidney function to help link your two topics here. A paragraph would be plenty to highlight current evidence regarding various antipsychotics and their influence on eGFR, CrCl, etc

Response: Thank you for this valuable suggestion. We have added the sentences in the introduction section discussing the current evidence regarding various antipsychotics and their influence on kidney function.

Introduction (page 4-5):

Several studies have investigated the effects of antipsychotic medications on kidney function with mixed results. While some retrospective studies suggest that atypical antipsychotics, including risperidone, may not increase the risk of acute kidney injury compared to typical antipsychotics,[14, 15] other population-based studies have reported increased risks in elderly patients.[16] The mechanisms underlying these effects remain unclear, but may involve direct nephrotoxic effects or indirect effects through metabolic changes.

2. there needs to be several rationale given for methodological choices: why that duration for risp treatment? why sample size even if you did

not perform a power calculation? why 140 as the cutoff for glucose which doesn't appear to rule out pre-diabetes? Why the significant instruction

and concern for the effect of the microbiome which isn't mentioned in the background? This becomes even more important since there is a sudden introduction of microbiome analysis in the methods. This should be in the background with rationale for its study.

Response: We have added detailed rationale for our methodological choices throughout the methods section in the revised manuscript.

Introduction (page 5):

….Therefore, we conducted a pilot randomized controlled trial to investigate the impact of short-term risperidone administration on the amino acid profile, kidney function, and glucose metabolism in healthy adults. Microbiome analysis was included because gut microbiome can metabolize D-amino acids, and we aimed to control for potential confounding effects of microbial D-amino acid metabolizm on our study endpoints.

Methods, Study design and participants (page 6):

….The glucose cutoff was chosen to exclude participants with overt diabetes while including those with potential insulin resistance, as measured by HOMA-IR.

….Sample size calculation was not performed due to the absence of prior data on effect size. This pilot study was designed to generate preliminary data for future power calculations.

Methods, Interventions (page 6):

….The lowest available dosage of 0.5 mg/day was chosen to prioritize participant safety. The 4-day treatment duration was chosen based on pharmacokinetic studies showing that steady-state levels are achieved within 3-4 days, allowing us to observe steady-state effects while minimizing exposure in healthy volunteers.

3. You have some instruction about dietary restrictions in this study, was protein intake a concern given its potential effects on renal labs?

Response: Thank you for this important consideration. We did not specifically restrict protein intake in this study, which is a limitation. We have added this to the limitations section in the revised manuscript.

Discussion (page 15):

…. Fourth, we did not specifically control for protein intake during the study period, which may have influenced renal laboratory parameters and represents a limitation of our study design.

4. The focus of the background is on renal function but the primary endpoint was HOMA-IR? Is it possible to explain why this is the primary

endpoint?

Response: We have clarified why HOMA-IR was chosen as the primary endpoint in the methods section.

Methods, Study endpoints (page 8):

The primary study endpoint was the mean change in HOMA-IR from baseline to 5 days between groups. HOMA-IR was selected as the primary endpoint because risperidone has the potential to increase glucose levels and insulin resistance, and we first aimed to assess the metabolic safety profile of risperidone in healthy individuals.

5. How and when were fecal samples collected for microbiome analysis?

Response: We have added detailed information about fecal sample collection procedures in the methods section.

Methods, Laboratory measurements (page 7):

Intestinal bacterial flora analysis was conducted based on 16S ribosomal ribonucleic acid (rRNA) sequencing of fecal samples using QIIME2 (version 2021.2) by Takara Bio’s Biomedical Center (Shiga, Japan). Fecal samples were collected by participants at home using a collection kit (FS-0017, TechnoSuruga Lab. Co., Ltd., Shizuoka, Japan) at baseline and on day 5.

6. Did between group comparisons of mean changes correct for baseline values?

Response: We did not adjust for baseline values in our between-group comparisons. The baseline characteristics were generally comparable between the two groups, which reduced the necessity for baseline adjustment.

7. For the demographic/baseline comparisons between the groups, can you put statistical analyses to these statements of being comparable and the BMI "likely" being different? Were any identified differences controlled for in a sensitivity analysis to assess for potential effects on primary

and secondary outcomes. Perhaps this should be included in the statistics of table 1?

Response: We have added statistical analyses to the demographic comparisons in Table 1 as mentioned in response to Reviewer 1, comment 1.

8. Table 1 footnote should explain what data is (%, means, s.d., etc).

Response: We have expanded the Table 1 footnote to explain the data presentation format.

Table 1 footnote (page 9-10):

Data are described as mean (standard deviation) for continuous variables or n (%) for categorical variables.

9. Is it possible to utilize the supplementary material to provide in-depth data of your microbiome analysis? The presentation of

microbiome results is underwhelming given the breadth of data generally obtained from microbiome analysis.

Response: We agree that the microbiome analysis presentation could be enhanced and have added more detailed microbiome results to the supplementary material.

Results, Effects of risperidone on the fecal microbiome (page 12):

…. Phylum-level analysis revealed that both groups were dominated by Firmicutes and Bacteroidetes, accounting for over 80% of the microbiome composition at baseline and no significant changes were observed in the composition at day 5 in either group (S2 Fig and S3 Table).

10. The discussion is lacking in depth and detail, especially as it regards the microbiome analysis, why your renal findings might contradict that

of other studies.

Response: We have substantially expanded the discussion section to provide more depth and detail, particularly regarding microbiome analysis and potential mechanisms explaining our findings.

Discussion (page 13-14):

…. Although fecal d-serine levels were not measured in the present study, we aimed to minimize confounding mediated dietary factors through the restriction of the intake of fermented foods, antibiotics, and lactic acid supplements during the study period. In addition, the lack of significant changes in gut microbiome diversity during the short treatment period suggests that the increases in D-serine levels were primarily due to direct enzyme inhibition rather than microbial effects. These findings ensured that the observed changes in d-serine were attributable to risperidone administration rather than the impact of dietary or gut microbiota-related factors.

….However, a population-based cohort study reported an increased risk of acute kidney injury-related hospitalizations in elderly patients treated with atypical antipsychotics, including risperidone.[16] The absence of acute adverse effects on kidney function in healthy participants in our study contrasts with some previous reports of antipsychotic-associated nephrotoxicity, which may be related to our study population (healthy individuals vs. patients with comorbidities) or the short duration of exposure. These discrepancies underscore the need for further studies to elucidate the kidney safety profile of risperidone particularly in vulnerable populations such as elderly individuals and patients with CKD.

---

## [Decision Letter · Decision Letter 1]

3 Nov 2025

Effects of risperidone on amino acid metabolism, glucose, and kidney function in healthy adults: A pilot randomized controlled trial

PONE-D-25-19801R1

Dear Dr. Iwata,

We’re pleased to inform you that your manuscript has been judged scientifically suitable for publication and will be formally accepted for publication once it meets all outstanding technical requirements.

Kind regards,

Kuo-Cherh Huang

Academic Editor

PLOS ONE

Additional Editor Comments (optional):

Dear Drs. Oshima and Iwata,

Basically, the reviewers were satisfied with your responses to prior review comments and the quality of the revised manuscript. Having said that, Reviewer 3 had a remaining suggestion for your work as follows: “Because outcomes were assessed at multiple time points (baseline, day 5, day 9), please indicate whether repeated-measures analysis or paired testing was considered to account for within-subject correlations.” Please address the issue in the final version of your manuscript. Thank you.

Kuo-Cherh Huang

Academic Editor

Reviewers' comments:

Reviewer's Responses to Questions

**Comments to the Author**

Reviewer #1: All comments have been addressed

Reviewer #3: All comments have been addressed

2. Is the manuscript technically sound, and do the data support the conclusions?

Reviewer #1: (No Response)

Reviewer #3: Yes

3. Has the statistical analysis been performed appropriately and rigorously?

Reviewer #1: (No Response)

Reviewer #3: Yes

4. Have the authors made all data underlying the findings in their manuscript fully available?

Reviewer #1: (No Response)

Reviewer #3: Yes

5. Is the manuscript presented in an intelligible fashion and written in standard English?

Reviewer #1: (No Response)

Reviewer #3: Yes

Reviewer #1: (No Response)

Reviewer #3: Because outcomes were assessed at multiple time points (baseline, day 5, day 9), please indicate whether repeated-measures analysis or paired testing was considered to account for within-subject correlations.

**Do you want your identity to be public for this peer review?** For information about this choice, including consent withdrawal, please see our Privacy Policy

Reviewer #1: No

Reviewer #3: No

---

## [Editor Report · Acceptance letter]

PONE-D-25-19801R1

PLOS ONE

Dear Dr. Iwata,

I'm pleased to inform you that your manuscript has been deemed suitable for publication in PLOS ONE. Congratulations! Your manuscript is now being handed over to our production team.

Kind regards,

on behalf of

Dr. Kuo-Cherh Huang

Academic Editor

PLOS ONE